# Diversifying Evolution of the Ubiquitin-26S Proteasome System in Brassicaceae and Poaceae

**DOI:** 10.3390/ijms20133226

**Published:** 2019-06-30

**Authors:** Zhihua Hua, Peifeng Yu

**Affiliations:** 1Department of Environmental and Plant Biology, Ohio University, Athens, OH 45701, USA; 2Interdisciplinary Program in Molecular and Cellular Biology, Ohio University, Athens, OH 45701, USA

**Keywords:** the ubiquitin-26S proteasome system, evolution, Brassicaceae, Poaceae

## Abstract

Genome amplification and sequence divergence provides raw materials to allow organismal adaptation. This is exemplified by the large expansion of the ubiquitin-26S proteasome system (UPS) in land plants, which primarily rely on intracellular signaling and biochemical metabolism to combat biotic and abiotic stresses. While a handful of functional genomic studies have demonstrated the adaptive role of the UPS in plant growth and development, many UPS members remain unknown. In this work, we applied a comparative genomic study to address the functional divergence of the UPS at a systematic level. We first used a closing-target-trimming annotation approach to identify most, if not all, UPS members in six species from each of two evolutionarily distant plant families, Brassicaceae and Poaceae. To reduce age-related errors, the two groups of species were selected based on their similar chronological order of speciation. Through size comparison, chronological expansion inference, evolutionary selection analyses, duplication mechanism prediction, and functional domain enrichment assays, we discovered significant diversities within the UPS, particularly between members from its three largest ubiquitin ligase gene families, the *F-box* (*FBX*), the *Really Interesting New Gene* (*RING*), and the *Bric-a-Brac/Tramtrack/Broad Complex* (*BTB*) families, between Brassicaceae and Poaceae. Uncovering independent *Arabidopsis* and *Oryza* genus–specific subclades of the 26S proteasome subunits from a comprehensive phylogenetic analysis further supported a diversifying evolutionary model of the UPS in these two genera, confirming its role in plant adaptation.

## 1. Introduction

To adapt to the ever-changing environmental conditions, all life forms have developed various kinds of systems through genome evolution. While animals learned to migrate away from unfavorable environments, the sessile life style of plants does not provide this advantage. Plants can experience much harsher and more variable living conditions compared to other organisms, particularly after their ancestral forms colonized the land area, where they began to suffer drought, salinity, light or adverse temperature stresses, as well as attacks by herbivores, pests and pathogens. To enable their dominance of large areas of Earth, plants have developed diverse strategies that have been encoded in their genomes throughout the past ~500 million years of evolution [1]. Human behavior has severely impacted much of the environment, potentially adding more stresses onto plants and impeding their survival. Learning the kinds of strategies that the plants utilize, and how they established these strategies in their genomes, will provide foundations for developing approaches to maintaining a sustainable agricultural production and a healthy living environment.

Throughout evolution, plants have learned how to regulate thousands of biochemical reactions and maintain a relatively stable internal environment by sensing, transferring, and responding to numerous environmental cues. Among the various regulatory machineries, the ubiquitin (Ub)-26S proteasome system (UPS) is of immense importance to plants, as evidenced by its multitude of roles that have been discovered in the past two decades of plant functional genomic studies [2,3,4,5,6,7,8,9]. The biochemical role of the UPS can be split into two stages: protein ubiquitylation by the ubiquitylation system, followed by subsequent degradation of ubiquitylated proteins by the 26S proteasome. Protein ubiquitylation involves three enzymes, Ub-activating (E1), -conjugating (E2), and -ligating (E3) enzymes, that catalyze the activation, conjugation, and ligation, respectively, of a 76 amino acid (aa) peptide, called Ub. Through ubiquitylation, the Ub moiety is covalently linked to the ε-amino group of a lysine (K) residue either on a ubiquitylation substrate or a preceding Ub molecule (acceptor) that has been previously attached to the substrate. In many cases, protein ubiquitylation yields chains containing multiple Ubs. Because there are seven K residues (K6, K11, K27, K29, K33, K48, and K63) on Ub and each K residue can accept a Ub donor to extend the poly-Ub chain, diverse topologies of the poly-Ub chain can occur [10]. In most cases, only when substrates are attached with homotypic K48-linked chains will they be recognized by the 26S proteasome and degraded [10]. The ubiquitylation and degradation processes are thus intimately interconnected. 

The UPS-mediated selective protein degradation allows the plants to rapidly turn biochemical reactions on or off, making it ideal for sensing and responding to environmental stresses. The UPS impacts nearly all aspects of plant life cycle, including hormone signaling, circadian rhythms, reproductive development, the cell cycle, and innate immunity [2]. Not surprisingly, many genomic studies have demonstrated that the UPS has dramatically expanded in plant genomes [11,12,13,14,15,16,17,18]. It has been estimated that as much as ~6% of the proteome in *Arabidopsis thaliana* (*Ath*) belongs to the UPS members [19]. If a roughly equal number of the UPS substrates are assumed [20,21,22], the size of the UPS-mediated regulatory pathways is overwhelming. Additionally, recent studies have shown cross-talk of the UPS with the other large proteolytic system, autophagy, in plants, as well as in other eukaryotic organisms [23,24,25,26,27]. The 26S proteasome itself is targeted for degradation by proteophagy via the binding of RPN10 to ATG8 through the Ub-interacting motif (UIM) and UIM-docking site (UDS), respectively [23,28].

Thanks to the advances in both genome and proteome sequencing technologies, we now can appreciate the breadth and depth of the UPS in regulating plant growth and development. However, our understanding of its function is still rudimental. For example, the functions of many ubiquitin E3 ligases have not been characterized [2]; many ubiquitylation substrates identified by proteomics-based liquid chromatography-tandem mass spectrometry (LC-MS/MS) require further confirmation since the majority does not possess a clear ubiquitylation footprint [21]; the lack of clear relationship between the size of the UPS and plant behavior hinders the identification of key regulatory members [11]; and the evolutionary mechanism driving the dramatic expansion of the UPS in plants remains unclear [2]. To address these questions, in addition to developing advanced experimental technologies, bioinformatics-based genomic sequence analyses have helped shed light on the underpinning molecular mechanism of the UPS. In this work, by taking advantage of a recently accomplished genome sequencing project on wild rice species [29], we systematically analyzed and compared most, if not all, UPS families in two evolutionarily distant plant families, Brassicaceae and Poaceae. To the best of our knowledge, we believe this is the first comprehensive genomic study of the entire UPS across multiple genomes. Our results demonstrate striking differences in UPS compositions between these two plant families, suggestive of diversifying evolution.

## 2. Results

### 2.1. Re-Annotation of UPS Gene Families

The UPS is composed of many constituent parts encoded within plant genomes. E1s, E2s, and E3s form the basic framework of the ubiquitylation system, while the 26S proteasome is also diverse in its composition (Appendix A). Several previous studies, including ours, have focused on the evolutionary study and duplication analysis of single or multiple families within this system [11,12,13,15,17,18]. However, it remains challenging to systematically analyze the whole system due to its large scale that hinders the finding of its members.

Our recent work in developing a highly automatic annotation program, called closing-target-trimming (CTT), allowed us to overcome this annotation limit [30]. Indeed, our previous studies have already applied this gene annotation algorithm to demonstrate that the annotation quality differs significantly among different genome projects, and that CTT can assist in finding most, if not all, members of any gene family in any genome [11,18,30]. Since each family in the UPS carries a characteristic protein–protein interaction domain feature (Appendix A), we used CTT and re-annotated the UPS members, including E1s, E2s, five E3 groups containing seven families, and both the regulatory and core particle families in the 26S proteasome [2,31,32,33] (Appendix A; Section 4). To understand the evolutionary pattern and functional divergence of the UPS in plants, we compared these members within two evolutionary distantly-related plant families, Brassicaceae and Poaceae. To reduce potential age and size-related errors, six species were selected in both families in a close chronological order of their split time in the evolutionary history [29,34,35,36]. In addition, the genomes of *Amborella trichopoda* (*Atr*) and *Zea mays* (*Zma*) were also analyzed as the common ancestor of both plant families [37] and control, respectively (Figure 1A, Appendix A).

We identified in total 21,299 loci including 1,062 pseudogenes that compose the aforementioned UPS groups in 14 genomes (Figure 1B, Appendix A). Although we are currently still missing annotation of the Cullin 4 (CUL4)-based and monomeric U-box-containing E3 ligases due to their lack of clear family seed sequences in Pfam (https://pfam.xfam.org), we feel that this large group of UPS data should reflect the overall UPS structure in these 14 plant species. We therefore hereafter designate the annotation result from 11 families as the UPS in each species. We discovered on average 1,521 UPS members in each species, from as low as 845 in *Atr* to as high as 2,204 in *Brassica rapa* (*Bra*). Among these, 3.1 to 25.9% of the total loci were newly annotated in this work (Figure 1B, Appendix A). 

The varying number of new loci discovered in this work seems to be related to the prior genome annotation quality. For example, only 45 out of 1,460 loci (3.1%) were new in the well-annotated *Ath* genome (Version 10). Comparing the UPS size with the genome size, we did not observe significant correlation (Figure 1C), likely due to different genome organization [38] and genomic-drift evolution of the UPS families [11]. To further examine the lack of this correlation, we annotated the 11 UPS families in *Zma* whose genome is 17.8-fold larger than *Ath*. Consistently, the number of protein sequences annotated in the *Zma* UPS was even smaller than that found in the *Ath* UPS (Appendix A). In contrast to the lack of size correlation between the UPS and the genome, a tight correlation was detected between the number of new peptide loci and the number of new pseudogenes annotated (Figure 1D). Therefore, the number of active UPS members is likely smaller than the size of the UPS.

This comprehensive annotation also allowed us to compare size variation of the UPS at the plant family level. We did not see significant differences in the size of the overall UPS between Brassicaceae and Poaceae (Figure 1B). However, after we split the UPS into 11 individual families and did pairwise comparisons, we found that these two plant families favor the expansion of different UPS gene families (Figure 2A). While the six Brassicaceae species have significantly more *anaphase promoting complex* (*APC*), proteasome *regulatory particle* (*RP*), and *F-box* (*FBX*) gene family members than the six Poaceae species, the latter group seems to encode more E1, Cullin, and Bric-a-Brac/Tramtrack/Broad Complex (BTB) family members than the former group (Wilcoxon rank sum test, *p* < 0.05; Figure 2B). 

To further address this plant family-level variation, we clustered the 13 species and 11 UPS families based on the number of loci with orthologous members (not orphan genes) in each UPS family of each species, and found a consistent trend (Figure 3A). Four species, *Bra*, *A. halleri* (*Aha*), *Capsella rubella* (*Cru*), and *A. lyrata* (*Aly*), all belonging to the Brassicaceae family, were clustered in one group that has high number of *FBX* loci and a small-sized *BTB* family. However, in another cluster, which contains *Boechera stricta* (*Bst*)*, Leersia perrieri* (*Lpe*)*, Oryza punctata* (*Opu*)*, Ath, Sorghum bicolor* (*Sbi*)*, O. sativa* (*Osa*), and *Brachypodium distachyon* (*Bdi*), small- or medium-sized *FBX* gene families and large-sized *BTB* gene families are evident (except for *Ath* and *Bst*, which are Brassicaceae species; Figure 3A). In addition, the number of *Cullin* and *E1* loci in this latter cluster was also higher than the former one. Since the species selected in both the Brassicaceae and Poaceae families have experienced similar evolutionary time periods (Figure 1A), the impact of between species variation in these two plant families is moderate. Therefore, our annotation concluded that there is considerable gene family size variation within the UPS in Brassicaceae and Poaceae.

### 2.2. Variance of Expansion History of the UPS Families

To understand why the UPS gene families varied in size in the two distant plant families, we assigned the UPS members in *Aly* and *Osa* at different age ranks based on the emerging time of their orthologous groups within Brassicaceae and Poaceae, respectively. We then analyzed the size expansion of the aforementioned 11 UPS families at each age rank in these two genomes (Figure 1A). Based on the number of orthologous genes at each age rank, the 11 UPS families can be clustered into three groups (Figure 3B). The *FBX* and the *Really Interesting New Gene* (*RING*) families form the largest group (Cluster I), followed by Cluster II that contained four medium size families, which are proteasome *core particles* (*CPs*), *RPs*, *BTBs*, and *E2s*. The remaining five gene families, *E1*, *APC*, *Homologous to the E6-AP Carboxyl Terminus* (*HECT*), *Cullin*, and *S-phase kinase-associated protein 1* (*Skp1*), were grouped into Cluster III. Among these three clusters, only the *FBX* and the *RING* family members and the *Osa BTB* family members are present in different age ranks, while the other families lack members in young age ranks, particularly those from Cluster III. Not surprisingly, the sizes of UPS families in Cluster III are small and some, such as *E1* and *Cullin*, are conserved and play essential roles in the ubiquitylation pathway [2,19]. Furthermore, most UPS family members are present in the aged groups (Age (Rank) 10 and A), suggesting that the UPS was largely expanded before flowering plants split. Interestingly, the size differences in UPS families between Brassicaceae and Poaceae is roughly reflected by the number variance of family members in these two aged groups. For example, consistent with the higher number of *FBX*, *RP*, and *APC* members in Brassicaceae than that in Poaceae, members from Ages 10 and A are also more enriched in *Aly* than that in *Osa*. Conversely, *Osa* encodes more *E1* and *Cullin* loci than *Aly* within these two age-ranks as seen at the corresponding plant family level (Figure 2B and Figure 3B). However, young duplicated *BTB* members seem to contribute more significantly in expanding the size of the *BTB* gene family in *Osa* (Figure 3B), suggesting the presence of differential expanding mechanisms.

### 2.3. Differential Evolutionary Selection

It is commonly agreed that the lineage-specific/young duplicated genes play adaptive roles. Recently studies also demonstrated a high rate of relaxed negative selection in lineage-specific genes [11,29,39]. To examine whether similar trends of evolutionary selection are present in the UPS, we calculated mutation rates of age-ranked UPS genes in *Aly* and *Osa* based on their orthologous pairs in *Ath* and *Obr*, respectively, within each rank (Figure 4 and Appendix A). Both *Aly*–*Ath* and *Osa*–*Obr* species pairs split in ~13–15 million years ago (Figure 1A). Therefore, the age-related error in evolutionary comparison is negligible. 

Consistent with a previous study in rice genomes [29], we found a significant negative correlation of *K_a_* (nonsynonymous substitutions per non-synonymous site) and *K_a_/K_s_* values, and slight differences of *K_s_* (synonymous substitutions per synonymous site) values with age ranks in the *RING* family in both *Aly* and *Osa* genomes (not including Age A) (Figure 4A, Appendix A), suggesting that the *RING* family members are under relaxed negative selection within both Brassicaceae and Poaceae lineages. However, many of these members may have been duplicated early then later lost in other species, resulting in either negative correlations of *K_a_* or slight differences of *K_s_* among different age groups (Figure 4A, Appendix A). Although a negative correlation of *K_a_/K_s_* values for *FBX* members with age is observed in *Aly*, their *K_a_* and *K_s_* values show a significant positive correlation with age (not including Age A) (Figure 4B, Appendix A), suggesting that the younger the *FBX* genes the greater the strength of their relaxed negative purifying selection in Brassicaceae, and that the FBX gene family expanded in a different way compared to the *RING* family. Consistent with this hypothesis and the small-scale expansion of young *FBX* genes in *Osa* (Figure 3B), we did not detect any significant correlations of evolutionary selection regimes with age in the *Osa FBX* family. For the remaining UPS families, no such correlations are observed, except that a negative correlation of *K_a_/K_s_* values with age is identified in the *Aly Skp1* gene family, suggesting that a co-evolutionary process is present between the *Skp1* and the *FBX* gene families (Appendix A, Appendix A). 

Despite the correlation variance of evolutionary regimes with age between both UPS families and plant families, we also see a significant variance of selection strengths between *Aly* and *Osa* (Figure 4). For example, the *K_a_/K_s_* values of *Aly FBX* genes and *RING* genes are significantly higher than those of the corresponding family members in *Osa*. By contrast, the *K_a_* and *K_s_* values of these two families are significantly lower in *Aly* than those in *Osa*. Significantly higher *K_a_/K_s_* values for *BTB* genes were also detected in *Osa* than those in *Aly*. Collectively, these data suggest that the three large UPS gene families, *FBX*, *RING*, and *BTB*, are under diversifying evolutionary processes in Brassicaceae and Poaceae, which are represented by the age-ranked orthologous UPS genes in *Aly* and *Osa,* respectively.

### 2.4. Relaxed Negative Selection Resulted in Neutralization

Relaxed negative selection may indicate a trend of neutralization. To test this hypothesis, we examined the neutral evolution of a UPS gene using the same approach employed previously to identify neutral changes of the *Arabidopsis Skp1-like (ASK)* genes [41]. By comparing the likelihood ratio obtained from two runs of codeml program with both fixed and free *K_a_*/*K_s_* ratios from the PAML4 package [40], we identified in total 113 out of 987 UPS genes (11.4%) that are under neutral changes in *Aly*. Not surprisingly, the rate of neutrally evolving genes in *Aly* is tightly negatively correlated with age (Spearman’s test, *p* = −1, *p* < $2.2\times 10^{-16}$), which is consistent with the trend of *K_a_/K_s_* values in the two largest UPS families, the *FBX* and *RING* families. Indeed, we detected a significant proportion of *FBX* and *RING* members that are under neutral changes, particularly in Ages seven to 10 (total average 29.6%, Figure 5A). However, only a small proportion of UPS genes, including the *FBX* and *RING* members, were detected under neutral changes in *Osa*, and the number of these neutrally evolving genes does not correlate with age (Figure 5A). 

Since there are 329 and 151 *Aly* and *Osa* UPS genes, respectively, that do not strictly follow the age rank but share orthologous members with *Ath* and *Obr*, respectively, we asked whether more neutrally evolving UPS genes are also present in *Aly* than in *Osa*. We tested the neutrality of *Aly*–*Ath* and *Osa*–*Obr* orthologous pairs that are presented in two, three, four, and five species, but do not follow the age rank, in Brassicaceae and Poaceae, respectively. As a comparison, the orthologous pairs that are conserved in flowering plants were also examined (present in five species within the family plus *Atr*). Similarly, we also detected a higher rate of gene neutralization in *Aly* than that in *Osa* (on average 31.4% and 2.3% for orthologous members within Brassicaceae and Poaceaea, respectively; Figure 5B). Although no neutrally evolving genes were detected in the conserved group of age-ranked members, we found that one out of 42 (2.4%) ancient *Aly*–*Ath* pairs were under neutral changes, suggesting that not all ancient UPS members are under strong functional constraints. However, we only observed one out of 11 (9.1%) young *Osa* UPS members (only present in *Osa* and *Obr*) in this non-age-ranked group was under neutral changes. Therefore, this neutral evolutionary analysis suggested that a significant proportion of Brassicaceae UPS members, which are represented by *Aly*–*Ath* orthologous pairs, are under neutral changes, i.e., low functional constraints. These neutrally evolving members are primarily from the *FBX* and the *RING* families,

### 2.5. Contribution of Recent Duplications to the Size Expansion of UPS Families

Tandem duplications and retrotranspostion have been found to play a key role in the recent expansion of the *ASK* (*Skp1*) family in *Ath* and the *Ubiquiton* family in 50 plant genomes [18,41], both of which belong to the ubiquitylation system. We next asked whether these two duplication mechanisms have also contributed to the differential expansion and diverse selection patterns of the UPS gene families across Brassicaceae and Poaceae species. 

Using a similar approach, we identified the loci that are tandemly connected or lack introns, and assigned them as tandem and retrotransposed loci, respectively (Figure 6). Consistent with the large sizes of the UPS, 77.1, 58.1, and 16.9% of *Aly FBX*, *RING*, and *BTB* members, and 62.6, 45.3, and 62.2% of *Osa FBX*, *RING*, and *BTB* members, respectively, are predicted to be tandemly duplicated. While the *FBX* and the *RING* families contain more tandemly duplicated genes in *Aly* than do they in *Osa*, the latter encodes more tandemly duplicated *BTB* members than the former (Fisher’s exact test, *p* < $1\times 10^{-5}$, Figure 6A), suggesting a significant contribution of tandem duplication in expanding these three large UPS gene families. In addition to the biased contribution of tandem duplication in expanding UPS families between *Aly* and *Osa*, we also discovered that more *FBX* genes arose from tandem duplications than *RING* genes in both species (Figure 6A).

The number of intronless UPS members is less than that of tandemly duplicated genes in *Aly* and *Osa*. However, the *FBX* and *RING* families still contains a high number of intronless genes in both *Aly* (44.0 and 35.4%, respectively) and *Osa* (39.5% and 41.5%, respectively, Figure 6B). In addition, the *Osa BTB* family contains a significantly higher number of intronless members than the *Aly BTB* family (47.4% vs. 9.1%, respectively, Figure 6B), indicating a potential role of retrotranspostion in expanding the *Osa* (Poaceae) *BTB* genes.

### 2.6. Functional Variance of the UPS Gene Families between Aly and Osa

In addition to their family-attributing domain feature, many UPS family members carry additional protein–protein interaction domains. For example, all the Ub E3 ligase members carry at least one substrate binding domain. To understand whether differential size expansion and selection may have diversified the functions of the UPS in Brassicaceae and Poaceae, we annotated the protein-protein interaction domains (E-value < 1) present in all 20,237 UPS protein sequences in Appendix A and compared their enrichment between Brassicaceae and Poaceae. In total, we identified 19 and nine different protein–protein interaction domains that are significantly enriched in Brassicaceae and Poaceae, respectively (Figure 7). 

Not surprisingly, these varied UPS members are primarily from the three largest E3 ligase families, the FBX, RING, and BTB families. While there are two protein-protein interaction domains each from the BTB family that are preferentially enriched in Brassicaceae (Arm and NPH) and Poaceae (BACK and MATH) (Figure 7A,B), all four different types of overrepresented protein-protein interaction domains in the FBX family, FBA, Kelch, LRR, and PP2, are exclusively enriched in the Brassicaceae species (Student’s t-test, *p* < 0.01) (Figure 7A). There are a broad range of protein-protein interaction domains from the RING family that are significantly enriched in Brassicaceae and Poaceae (11 and 6, respectively, Figure 7), further highlighting functional divergence of the UPS in these two plant families.

### 2.7. Diversifying Evolution of the 26S Proteasome in the Genera of Arabidopsis and Oryza

Through protein–protein interaction domain analysis, we noticed that in addition to the three large E3 ligase families, both CP and RP protein families from the 26S proteasome also varied their domain compositions in Brassicaceae and Poaceae (Figure 7). The 26S proteasome acts as a protein degradation machinery through the recognition of poly-ubiquitylated proteins by multiple RP subunits, followed by protein catabolism in its central CP compartment [19,42,43]. However, it was unknown whether the 26S proteasome itself also contributes to the UPS diversity. The differential enrichment of protein-protein interaction domains in these two distant plant families suggests that the function of 26S proteasome is also diversified in plants. To further address this question, we carried out a phylogenetic analysis. We retrieved the CP and RP family members from three species each in the *Arabidopsis* and *Oryza* genera, which are in Brassicaceae and Poaceae, respectively. Although these two genera are phylogenetically distant, the species selected were split in a similar short age rank so that within-genus sequence variation is minimized. We further applied an improved phylogenetic analysis for both RP and CP families, which included the use of Trimal to automatically remove poorly aligned regions [44], and RaxML for generating a maximum likelihood phylogenetic tree [45] (Figure 8, Appendix A). To identify which isoforms are more diverged, we also included the RP and CP family members of the ancestral flowering species, *Atr*, in the phylogenetic analysis.

The annotation of RP family members is more challenging than the CP family due to both sequence divergence between family members and sequence homologies between RP family members and those from other protein complexes, such as eukaryotic translation initiation factor 3 (eIF3) complex and the COP9 signalosome (CSN) complex [46,47]. Indeed, regulatory particle non-ATPase 12 (RPN12) a and b subunits contain only one predictable protein-protein interaction domain, CSN8_PSD8_EIF3K, that is also presented in both eIF3 and CSN members. Therefore, in this work, RPN12 is not included for the analysis. For the same reason, RPN11, RPN9B, and regulatory particle triple-A ATPase 1A (RPT1A) were not analyzed. Except for these four members, the remaining 10 RPT members and 14 RPN members from the well characterized *Ath* RP family were all identified, in addition to the finding of a new RPN1-like protein (AT4G08140; Figure 8A and Appendix A), suggesting the efficiency and accuracy of our highly-automatic CTT annotation program [30]. Within the RP phylogenetic tree, we surprisingly noticed a significant diversity present between the two genera studied. First, the *Oryza* species appear to lack the members of RPN8, RPN9A, and RPT1B. Second, the *Oryza* species have one additional copy of RPN1 and RPT3 compared to the *Arabidopsis*. Third, within all subunits except for RPT4, the members from *Oryza* and *Arabidopsis* genera form an independent subclade, which is supported with strong statistical significance from 1,000 bootstrap analyses. Fourth, in 7 out of 9 RPN clades, the members from the ancestral flowering plant genome, *Atr*, do not fall in subclades of either genus. In addition, two clades, which contain *Ath* eIF3e and COP11, were mixed with RP subunits. Taken together, our data suggest that the RP members are highly diversified, and that their functions are significantly diverged from the common ancestor genome, *Atr*, in both *Arabidopsis* and *Oryza* genera.

Using a clear family domain feature of the CP family (Proteasome domain, Appendix A), we were able to identify exactly the same set of the *Ath* CP members as previously characterized using affinity purification and LC-MS/MS characterization (Figure 8B and Appendix A; [42]), further demonstrating the accuracy of our bioinformatics prediction. Interestingly, the CP phylogenetic tree not only resolved independent CP subunits but also divided each subunit into *Arabidopsis* and *Oryza* subclades with strong statistical evidence from 1,000 bootstrap analyses. Unlike the RP family, each CP subunit has members from both genera, suggesting that they have stronger functional constraints than RP members. However, the number of isoforms of each subunit varies between *Arabidopsis* and *Oryza* genera. For example, the *Oryza* has only one isoform for PAF, PAE, PBB, PBD, and PBE subunits, while two of each are present in the *Arabidopsis* genus. Conversely, the three *Oryza* genomes encode two isoforms each for PAG, PBA and PBF subunits while the *Arabidopsis* genus only contain one each in the three genomes analyzed. Similar to the RP family, only two *Atr* CP members are separately grouped within the *Arabidopsis* PBD and Poaceae PAB subclades. The remaining 23 *Atr* CP members are independent of the subclades formed by either genus. Therefore, even in the proteolytic central core compartment of the 26S proteasome, a functional diversification process has occurred in these two distant genera.

## 3. Discussion

Over the past two decades of functional genomic studies on the model plant, *Ath*, we have accumulated tremendous knowledge about how plant cells regulate gene expression to maintain their homeostasis in response to a changing environment. Among these, the role of the UPS has also been well appreciated [2,3,4,5,6,7,8,9]. However, due to its large scale and rapid evolving process, the functions of many members still remain elusive [2]. Our recent studies on the duplication mechanism of the ubiquitin and ubiquitin-like protein family discovered that the expression of the founding member of the UPS, Ub, is itself under species/lineage-specific regulation [18]. This raised a new intriguing question as to whether the UPS has been significantly diversified in distant plant lineages. In this study, through a comprehensive bioinformatics sequence analysis along with evolutionary selection and phylogenetic assays, we provided several lines of evidence demonstrating striking diversity of the UPS in the Brassicaceae and Poaceae families. These include i) size variation of the UPS families and subfamilies, particularly the three largest E3 ligase families, the FBX, the RING, and the BTB families (Figure 2B, Figure 3B, and Figure 7), ii) the strengths of evolutionary selection (Figure 4), iii) the rate of pseudogenization (Figure 5), iv) the recent duplication mechanisms (Figure 6), and v) phylogenetic variation (Figure 8). 

While the functions of many UPS members are not yet known, our research provided evolutionary evidence further demonstrating the complexity of this system. This complexity confirmed the adaptive roles of the UPS in combating the ever-changing environmental cues. Previous comparative genomic studies on five *Oryza* AA genomes revealed rapid diversification in association with rice adaption. Gene ontology (GO) analysis of in total 537 nonredundant positively selected genes revealed a statistically significant enrichment in response to developmental processes, particularly in flower development and embryogenesis, in addition to pathogen defense and disease resistance [48]. It has been shown that the UPS plays important roles in both plant reproduction and pathogen defense [2]. It will be interesting to further analyze whether the diversifying evolution of the UPS primarily contributes to the functional divergence in these two biological processes in Brassicaceae and Poaceae.

Due to the limited number of functionally-characterized UPS genes, it is not yet clear why the UPS, including that from other eukaryotic organisms, needs to develop several distinct E3 ligase families, which vary significantly in sizes (Figure 2B, Figure 3B and Figure 7). Whether this difference reflects their functional divergence and/or resulted from distinct gain-and-loss mechanisms has not been addressed. In this work, our data suggested that the size variance of the three largest E3 families is likely caused by differential expansion mechanisms given the following evidence. First, tandem duplication plays a more significantly role in expanding the *FBX* genes than in any other families in both *Aly* and *Osa* (Figure 6A), while intronless *BTB* genes (suggestive of a retrotransposition process) are significantly enriched in *Osa,* suggesting differential impacts of genome duplication mechanisms. Second, these differential impacts can be also reflected in orthologous sequence mutations. Mutation analysis on age-ranked *FBX* and *RING* orthologous pairs demonstrated an opposite correlation of *K_a_* and *K_s_* mutations with age in both *Aly* and *Osa*. While *K_a_* mutations may arise from adaptation or neutral changes, *K_s_* has been recognized to be relatively neutral, and can roughly serve as a molecular clock for a sequence [49]. The different trend of *K_s_* mutations upon the age rank in the *FBX* and *RING* families suggests distinct fate of family members in these two families. The young *FBX* genes with low *K_s_* values are consistent with their recently tandem duplication mechanism, which has not given enough time to accumulate mutations. However, the high *K_s_* values observed in the *RING* family members at the young age rank is not consistent with the tandem duplication model. One explanation would be due to gene loss. While ancient genome duplications increased the size of the *RING* families, differential gene loss resulted in contraction of the family at different stages. The ancient duplications may generate too many redundant *RING* copies, which resulted in their rapid loss. This model also fits the sudden increase or decrease of the *FBX* and the *RING* orthologous members, respectively, when Brassicaceae and Poaceae emerged. Interestingly, the number of *RING* family members is greater than two-fold of the *FBX* loci in the ancient UPS group (Group of Age A; Figure 3B). 

## 4. Materials and Methods

### 4.1. Data Resources and UPS Family Annotation

For each genome, the unmasked genomic sequence file, the generic feature format 3 (GFF3) file, and the annotated protein sequence file were downloaded from either Phytozome V12 (http://phytozome.jgi.doe.gov) or Ensembl Plants (http://plants.ensembl.org/index.html) (Appendix A). We applied a recently-developed CTT program to retrieve and reannotate most, if not all, UPS family members ([30,50]; https://github.com/hua-lab/ctt.git). The seed sequence files for the 11 UPS families were retrieved from Pfam 32 (https://pfam.xfam.org; Appendix A). Due to the presence of some protein-protein interaction domains in other protein families, we defined an RP family protein by one of the following domain combinations, Pro_isomerase, Prot_ATP_ID_OB, RPN1_RON2_N, RPN2_C, Rpn3_C + PCI, RPN5_C + PCI, RPN6_N + PCI, RPN7 + PCI, YfdX + PCI, MitMem_reg + Peptidase_M13_N, and VWA_2 +UIM (Appendix A). We also narrowed down the Cullin family members with the presence of both Cullin and Cullin_Nedd8 domains (Appendix A). We used the subfamilies in the SCOOP of RINGv predicted to be an E3 ligase on Pfam to annotate RING E3 members (https://pfam.xfam.org; [51]; Appendix A). Prok-RING_4 is excluded because it is found sporadically in bacteria [52]. All the protein–protein interaction domains were predicted based on HMMR3 search against the Pfam 32 database (E-value cutoff = 1).

### 4.2. Age Ranked Orthologous Group Analysis

The age rank is defined based on the split time of each genome according to previous studies [29,35]. To identify orthologous groups, the annotated members of each UPS family were subject to similarity analysis using OrthoMCL [53]. An inflation value of 1.5 was applied as in our previous studies on the Ubiquiton protein family [18]. The age ranked *Aly* orthologous pairs of each UPS family were defined based on their absence (0) and increasing presence in *Aha* (6), *Aha* + *Ath* (7), *Aha* + *Ath* + *Cru* (8), *Aha* + *Ath* + *Cru* + *Bst* (9), and *Aha* + *Ath* + *Cru* + *Bst* + *Bra* (10). A similar ranking was done for the orthologous pairs of the *Osa* UPS families. 

### 4.3. Clustering Analysis

Sequences were clustered using Heatmap.2 (dist method = “manhattan”, hclust method = “word.D”) in R (http://www.r-project.org) to demonstrate similar evolutionary process across the 13 genomes as described previously [18]. 

### 4.4. Detection of Recent Gene Duplications

Tandem genes were determined if two genes from the same subfamily were separated by 10 or fewer genes and were located within 300 kb. Intronless genes were identified based on the number of introns of each locus counted from GFF3 files.

### 4.5. Mutation Rate and Neutral Evolution Analysis

The *Aly*–*Ath* and *Osa*–*Obr* orthologous pairs at age ranks 7, 8, 9, 10, and A were resolved based on “Age Ranked Orthologous Group Analysis”. The coding sequence alignment of each pair was generated using the aligned protein sequences as a template in T-Coffee [54], and used as an input file for mutation rate analysis in a codeml program from the PAML4 package [40]. Two codeml runs were performed with the *K_a_*/*K_s_* ratio either fixed at 1 or free. Mutation rates, *K_a_* and *K_s_*, were determined in a codeml run when *K_a_*/*K_s_* ratio is set as free. The maximum likelihood (ML) values obtained from each codeml run (ML1 and ML2) were collected to obtain a likelihood ratio as LR  =  2(lnML1 − lnML2). According to the PAML4 package [40], we defined a UPS orthologous pair as being under neutral changes if LR is less than 2.71 (5% significance for *χ2* distribution with one degree of freedom).

### 4.6. Proteasome Sequence Alignment and Phylogenetic Analysis

We applied a similar sequence alignment approach for removing ambiguous alignment regions as we did in a previous phylogenetic study on the *ASK* genes [41]. Briefly, the RP and CP protein sequences were retrieved separately from three species each in both *Arabidopsis* and *Oryza* genera. In addition, the sequences of the corresponding family from the common ancestor genome of flowering plants, *Atr*, were also included. The combined protein sequences of each family were aligned using MUSCLE [55]. The resulting sequence alignment file was analyzed by Trimal to remove poorly aligned regions using an heuristic method (-automated1) [44]. The trimmed alignment was used to generate a maximum likelihood phylogenetic tree by RAxML (Version 8.1) with the PROTGAMMAJTT substitution model [45]. The statistical significance was evaluated with 1,000 bootstrap replicates using a rapid bootstrap analysis.

## Figures and Tables

**Figure 1 ijms-20-03226-f001:**
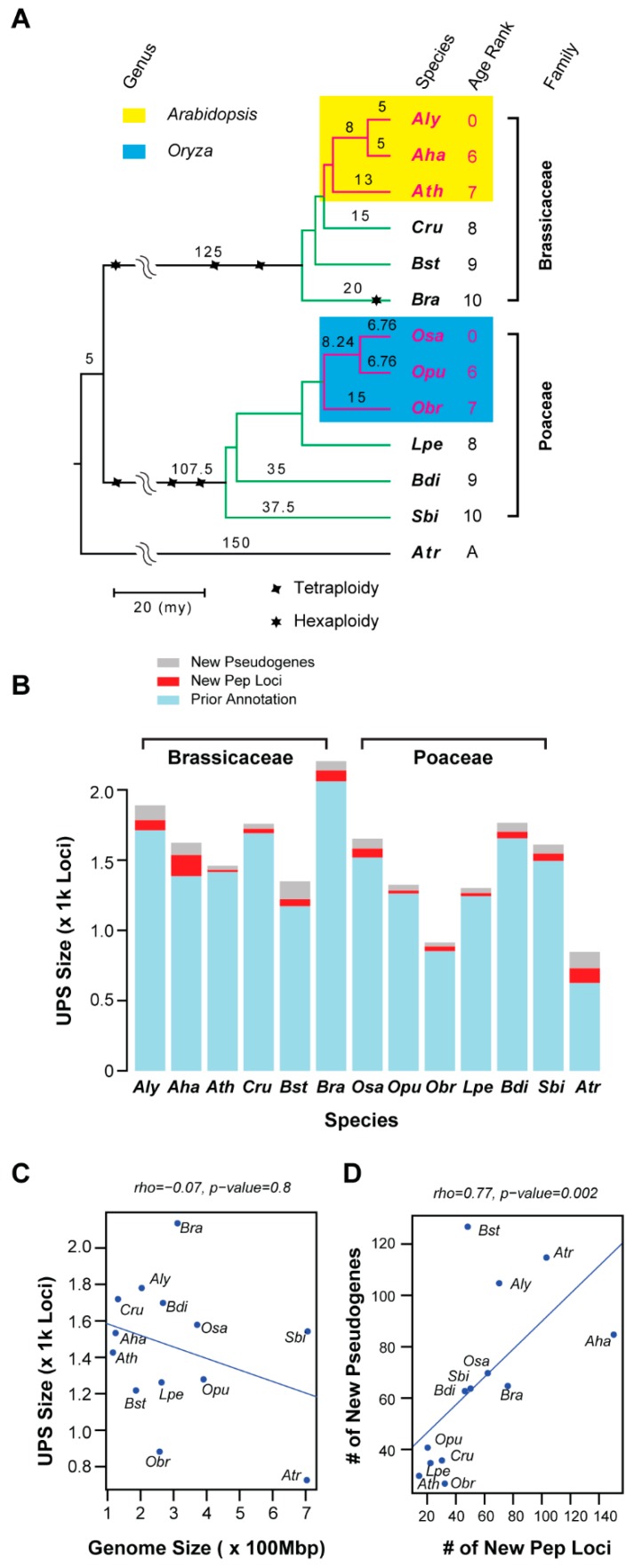
Annotation of the UPS members in Brassicaceae and Poaceae. (**A**) Phylogenetic relationship of 13 selected species. The tree topology and whole genome duplication events were inferred from literature [29,35]. The age rank is assigned for ordering purpose based on the split time of each species as defined in literature [29]. Species abbreviations in this panel and others are described in Appendix A. (**B**) Comparison of the UPS sizes across 13 plant species. (**C**) No correlation is present between the genome size and the UPS size in the 13 plant genomes analyzed. Mbp: million base pair. (**D**) A significant correlation is present between the number of new pseudogenes with the number of new peptide loci in the UPS group of the 13 plant genomes.

**Figure 2 ijms-20-03226-f002:**
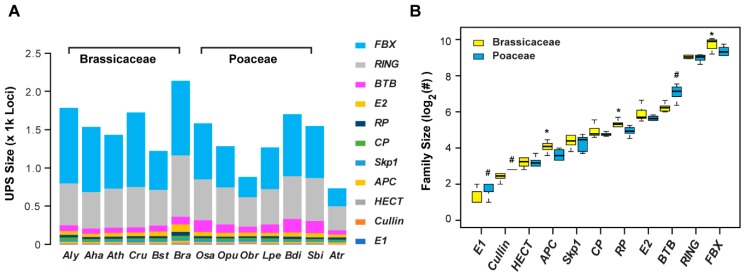
Varied size expansion of the UPS gene families in 13 plant genomes. (**A**) Size comparison of 11 UPS gene families in 13 plant genomes. A schematic diagram of 11 UPS gene families analyzed in this panel and others is described in Appendix A. (**B**) Unequal expansion of 11 UPS gene families in Brassicaceae and Poaceae. Single asterisks indicate that the size of the family is larger in Brassicaceae than that in Poaceae. Hash symbols dedicate a larger family size in Poaceae than in Brassicaceae. Statistical significance was performed based on Wilcoxon rank sum test (*p* < 0.05).

**Figure 3 ijms-20-03226-f003:**
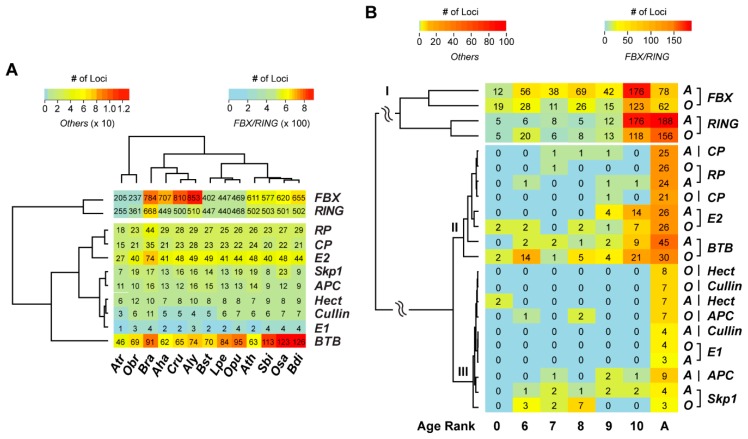
Size comparison of orthologous members across plant genomes revealed large, medium, and small groups of UPS families. (**A**) Clustering of OrthoMCL members reflected differential expansion of 11 UPS families across 13 plant genomes. (**B**) Age-ranked expansion of orthologous genes from 11 UPS families in *Aly* (*A*) and *Osa* (*O*) genomes revealed three major clusters. The number of orthologus genes is counted based on the orthologous groups present in each age rank. Orthologous group age ranking is described in Section 4.

**Figure 4 ijms-20-03226-f004:**
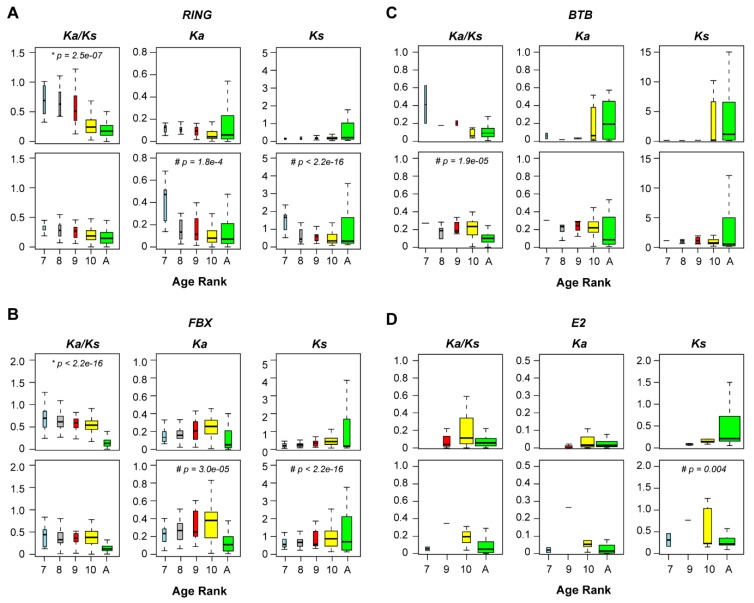
Evolutionary selection analysis of age-ranked orthologous genes from four large UPS gene families. Age ranked orthologous *Aly*–*Ath* (top panel) and *Osa*–*Obr* (bottom panel) pairs were retrieved as in Figure 3B and subject to codeml analysis in PAML4 [40] with a *K_a_/K_s_* free model. The values of *K_a_/K_s_*, *K_a_*, and *K_s_* of each pair of *RING* (**A**), *FBX* (**B**), *BTB* (**C**), and *E2* (**D**) orthologous genes were grouped and plotted using boxplot in R. Age rank is described in Figure 1A. Single asterisks and hash symbols indicate that the corresponding values in *Aly*–*Ath* orthologous pairs from ages seven to 10 are greater or less than that in *Osa–Obr* pairs within the same age ranks, respectively. Statistical significance was performed based on Wilcoxon rank sum test (*p* < 0.05). The width of each boxplot is proportional to the number of orthologous pairs.

**Figure 5 ijms-20-03226-f005:**
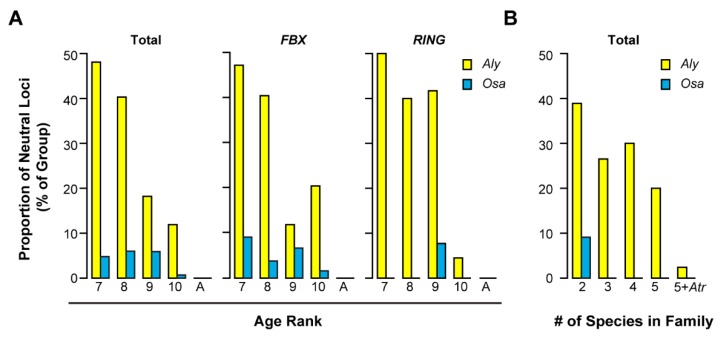
Neutrality test revealed a significant proportion of pseudogenes in *Aly*–*Ath* UPS orthologous pairs. (**A**) Age-ranked orthologous *Aly*–*Ath* and *Osa*–*Obr* pairs were retrieved as in Figure 3B and subjected to neutrality analysis as described in Section 4. The proportion of neutrally evolving loci was plotted against age ranks. (**B**) Non-age-ranked *Aly*–*Ath* and *Osa*–*Obr* pairs were retrieved based on the number of species present in the same orthologous group from Brassicaceae and Poaceae, respectively. Each pair was subjected to neutrality analysis as in (**A**). The proportion of neutrally evolving loci was plotted against the number of species present in each orthologous group.

**Figure 6 ijms-20-03226-f006:**
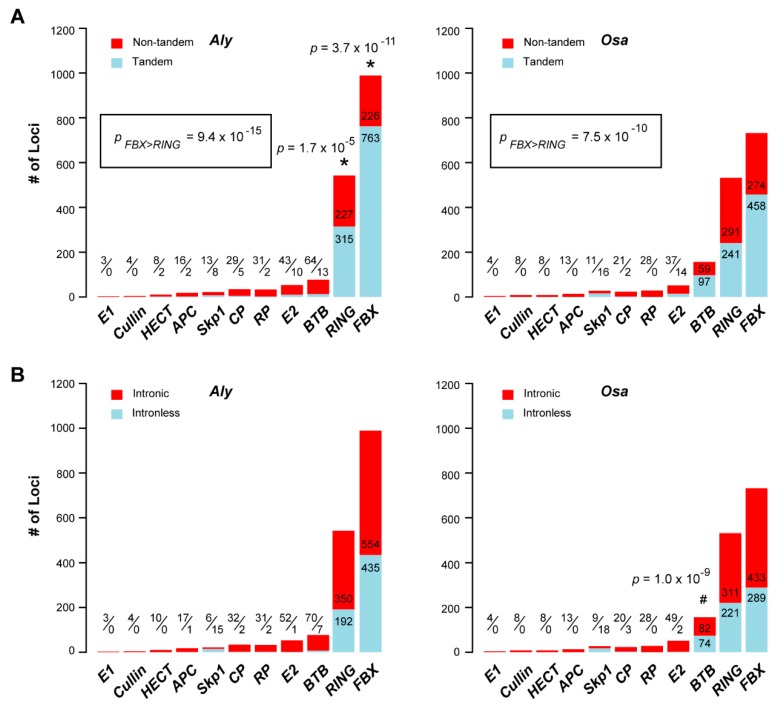
Comparison of the role of recent duplications in expanding the sizes of 11 UPS gene families in *Aly* and *Osa*. (**A**) Tandem duplication plays a more significant role on the expansion of the *FBX* and *RING* gene families in *Aly* than in *Osa*, and vice versa for the *BTB* family. In both genomes, more FBX genes were tandemly duplicated than *RING* members. The sizes of tandem and non-tandem loci are indicated either in the denominator and numerator of a fraction, respectively, or directly on the bar plot. (**B**) Significantly more intronless *BTB* loci were found in *Osa* than in *Aly* but not others. The sizes of intronless and intronic loci are indicated either in the denominator and numerator of a fraction, respectively, or directly in the bar plot. Single asterisks indicate a significant enrichment of tandem duplications in *Aly*. Hash symbols denote a significant enrichment of tandem duplications or intronless genes in *Osa*. Inset shows the enrichment comparison of tandem duplications between the *FBX* and the *RING* families in *Aly* and *Osa* genomes. Statistical significance was performed based on Fisher’s exact test and the *p* values are indicated.

**Figure 7 ijms-20-03226-f007:**
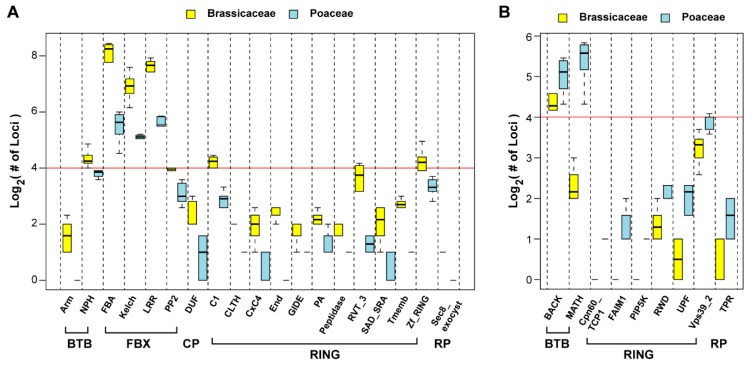
Domain enrichment assay revealed biased expansion of UPS subfamilies in Brassicaceae and Poaceae species analyzed. (**A**) Boxplots show 19 UPS subfamilies with a significantly more loci in Brassicaceae than in Poaceae. (**B**) Nine UPS subfamilies are shown to be enriched in Poaceae compared to Brassicaceae. Statistical significance was performed based on Student’s t-test (*p* < 0.01).

**Figure 8 ijms-20-03226-f008:**
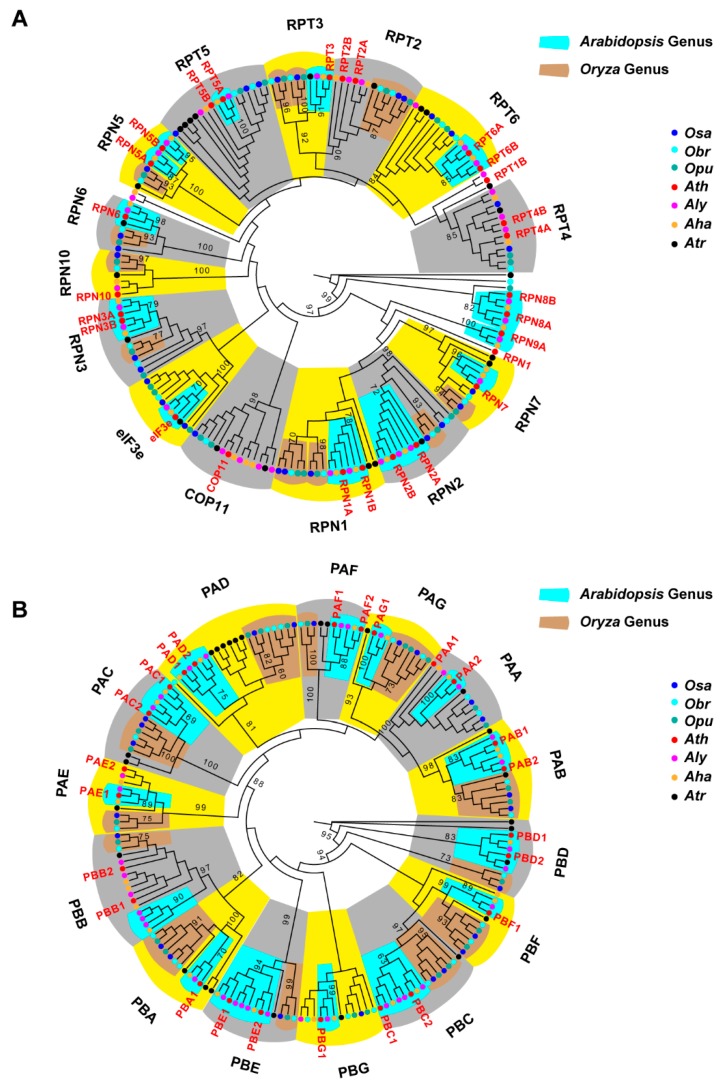
Phylogenetic analyses uncovered diversifying evolution of the 26S proteasome in *Arabidopsis* and *Oryza* genera. (**A**) Phylogenetic relationship of 9 RPN and 6 RPT subunits. (**B**) Phylogenetic relationship of the complete set of CP subunits. In each panel, the sequences of each member were annotated and subjected to a maximum likelihood phylogenetic analysis as described in Section 4. Detailed sequence IDs are described in Appendix A. The name of an *Ath* isoform is indicated in red. Subunit clades are alternately shaded with yellow and gray color. Statistical significance equal to or greater than 60% of 1,000 bootstrap re-samplings is indicated in a node that separates a significant clade. Significant clades from *Arabidopsis* or *Oryza* genus only are shaded with light blue or brown color, respectively.

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
