# Peer review of "Diversifying Evolution of the Ubiquitin-26S Proteasome System in Brassicaceae and Poaceae"

_ijms, 2019, doi:10.3390/ijms20133226_

Round 1
Reviewer 1 Report
In this paper, Hua et, al reported a comprehensive bioinformatics analyses of the ubiquitin-26S proteasome system (UPS) in the Brassicaceae and Poaceae genera. UPS is large and complex in plant kingdom. Its targets involve in many aspects of plant biology. Overall the authors did a good job.
1.The authors need to give more descriptions about the specie selection. Two agricultural important species, wheat and maize, are excluded from the analysis. I think because of their large and highly repetitive genomes and agricultural importance; it is even more interesting to include them in the analysis. The reason stated by the authors is “to reduce potential age and size-related errors”. Please give more explanations about these “potential errors”. If the authors want to only include species with similar genome sizes, then what is the point of correlating the sizes of genomes with the number of UPS detected in Figure 1C. I would argue adding large genomes like wheat and maize might make this figure panel more informative. Wheat and rice diverged from each other 50 MYA (million years ago). If Atr (150 MYA) and Sbi (37.5 MYA) are included, it is appropriate to include wheat and maize too.
2.About the age ranking in Figs 1A, 3, 4 and 5,
a.why Aly of Brassicaceae and Osa of Poaceae are ranked 0, and then the next rank is 6 (Aha and Opu). The difference is 6. While the differences between other ranks are 1. Do the authors use them as categorially or quantitatively?
b.Bra diverged from Cru 10 MYA and their ranks are 10 and 8, respectively. Sbi diverged from Lpe 37.5 MYA and their ranks are also 10 and 8. The divergence ages are very different, is it appropriate to give them the same rank?
3.The color of oryzease does not match the color used in the Figure 1A.
4.Fig 3A, it is better to label the individual numbers in the heatmap (same as Fig 3B to be consistent), because the different color distributions of the two colorbars could be visually misleading on the heatmap.
Author Response
Dear Editor,
In this resubmission, we have summarized below how we have addressed all the comments raised by the reviewer. The reviewer’s comments are copied verbatim in italics. Also, for those paragraphs of the reviewer’s reports that contain comments on unrelated issues, we have broken them down into separate items and addressed each issue separately. In this revision, we also carefully corrected some minor mistakes in Tables S3 and S4 and Figures 2 and 6 in our first submission that have been separately addressed at the end of this letter. These corrections did not change any conclusions made in our work. Please note that unless otherwise stated, all page and figure numbers refer to those of the revised manuscript.
We sincerely thank both referees for their constructive comments, and we believe that having addressed their comments, we have strengthened our manuscript. We hope that you and the referees find our revisions satisfactory.
Response to Reviewer 1
Comment 1:In this paper, Hua et, al reported a comprehensive bioinformatics analyses of the ubiquitin-26S proteasome system (UPS) in the Brassicaceae and Poaceae genera. UPS is large and complex in plant kingdom. Its targets involve in many aspects of plant biology. Overall the authors did a good job.
We appreciate the positive feedback from this reviewer.
Comment 2:1.The authors need to give more descriptions about the specie selection. Two agricultural important species, wheat and maize, are excluded from the analysis. I think because of their large and highly repetitive genomes and agricultural importance; it is even more interesting to include them in the analysis. The reason stated by the authors is “to reduce potential age and size-related errors”. Please give more explanations about these “potential errors”.
Our method of specie selection was inspired by the work from Stein et al. (Nature Genetics., 2018(50):285-296). Stein et al. did excellent genomic studies on the conservation, turnover and innovation of genes across the genus Oryza. Since the UPS is composed of a large group of rapidly evolving members, we sought to compare the differential evolutionary patterns in two distinct plant lineages, Brassicaceaeand Poaceae.
The present plant species have experienced different whole genome duplications, including polyploidy(e.g., Triticum aestivum(wheat) which is hexaploid) and recent whole genome duplications (e.g., Zea mays and Brassica rapa(Bra), which experienced a whole genome duplication within species). Such isolated whole genome duplication events may significantly skew the size of superfamilies. For example, among 13 species we analyzed, the UPS size of Brais obviously greater than the rest 12 species (Figure 1B). To reduce this bias, we intended to select the species that have experienced a similar number of whole genome duplication events. However, due to the limit of available genomes, we still kept Brafor the comparison.
Please see our response to Comment 4 regarding the data analysis of wheat and maize genomes.
Comment 3:If the authors want to only include species with similar genome sizes, then what is the point of correlating the sizes of genomes with the number of UPS detected in Figure 1C.
It was not our purpose to select genomes with similar sizes. As we stated in response to Comment 2, the species we selected have experienced a similar number of whole genome duplication events. Our previous genomic studies have demonstrated that many UPS superfamilies such as the F-boxsuperfamily experienced genomic-drift evolution, in which the small-scale duplication events (such as tandem duplications and retrotransposition duplications) play important roles in the expansion of superfamilies. Our discovery in this work showing no correlation between genomes with the number of UPS detected in Figure 1C further supported this hypothesis. Consistently, we discovered that both tandem and retrotransposition duplication events contributed significantly to the generation of the three largest UPS families, the F-box, the RING, and the BTB families (Figure 6). In this revision, we added one sentence to explain the possible contribution of genomic-drift evolution in the expansion of the UPS families (Lines 13-14, Page 7). Genomic-drift evolution was first presented by Masatoshi Nei to explain the lack of correlation between the size of olfactory superfamily and the complexity of chemosensory phenotype in vertebrates (Nozawa et al., PNAS 2007)
Comment 4:I would argue adding large genomes like wheat and maize might make this figure panel more informative. Wheat and rice diverged from each other 50 MYA (million years ago). If Atr (150 MYA) and Sbi (37.5 MYA) are included, it is appropriate to include wheat and maize too.
We agree with this comment. Adding more genomes may increase the statistical power of Figure 1C. However, wheat and maize genomes reflect outliers due to recent whole genome duplications. Since wheat genome is composed of three different sets of subgenomes, we don’t feel that it is comparable to the diploid genomes we analyzed. However, in this revision we added our new annotation of the UPS members from the maize genome as suggested by this reviewer. The result showed that the number of the maize UPS protein sequences is even smaller than that of the AthUPS although its genome is 17.8-fold larger than the Athgenome. This new data is added in Lines 14-21, Page 7. Accordingly, Tables S2, S3 and S4 and Files S1 and S2 were revised.
Comment 5: 2.About the age ranking in Figs 1A, 3, 4 and 5,
a.why Aly of Brassicaceae and Osa of Poaceae are ranked 0, and then the next rank is 6 (Aha and Opu). The difference is 6. While the differences between other ranks are 1. Do the authors use them as categorially or quantitatively?
The age rank is essentially used for ordering purpose (not for quantification). Each selected species is ranked based on its split time in the history. The number of ranks is selected to be consistent with the previous publication by Stein et al. (Nature Genetics., 2018(50):285-296). For example, among the six Poaceaespecies (Osa, Opu, Obr, Lpe, Bdi, Sbi) used in our work, their age ranks were assigned to be 0, 6, 7, 8, 9, and 10, respectively, by Stein et al (Nature Genetics, 2018(50):285-296). Since we selected the six Brassicaceaespecies based on their approximately similar emerging time period as the six Poaceaespecies, we assigned them with the same order of age ranks. In our original submission, we briefly described the method of how we assigned the age ranks in the section of “Age Ranked Orthologous Group Analysis” in “Materials and Methods”. In this revision, we added one sentence in the legend to Figure 1A, that reads “The age rank is assigned for ordering purpose based on the split time of each species as defined in literature {Stein, 2018 #28}.”. We hope now it is much clear.
Comment 6: b.Bra diverged from Cru 10 MYA and their ranks are 10 and 8, respectively. Sbi diverged from Lpe 37.5 MYA and their ranks are also 10 and 8. The divergence ages are very different, is it appropriate to give them the same rank?
Once again, the age rank is for ordering/trend purpose not for quantification. We have done our best to select two groups of species that split in a similar order in history from two distinct plant families, Brassicaceaeand Poaceae. Species selected from distinct evolutionary histories may complicate the comparison due to whole genome duplication and/or segmental duplication events. Please also see our responses to Comments 2, 3 and 5.
Comment 7: 3.The color of oryzease does not match the color used in the Figure 1A.
Thank you for carefully reading our manuscript. This mistake has been corrected in the revision.
Comment 8: 4.Fig 3A, it is better to label the individual numbers in the heatmap (same as Fig 3B to be consistent), because the different color distributions of the two colorbars could be visually misleading on the heatmap.
We appreciate this suggestion. The number of loci in each category has been added in the heatmap.
Additional Changes
1. The number of “Prior Annotation” in Table S3 was not correct. It was a sum of “Prior Annotation” and “New Pep” genes. We have corrected this mistake in this revision. Accordingly, we revised Figure 1B. This revision does not change any conclusions made in this work.
2. The number of loci in Table S4 was not correct. In our first submission, we calculated these numbers without removing differential splicing variants, which resulted in inconsistency between the number of loci listed in this table and the number of sequences collected in File S1 and S2. In this revision, we have carefully removed the splicing variants with short lengths from the same locus. The number of loci in the new Table S4 matches the number of sequences in Files S1 and S2 now. Accordingly, Figures 2A, 2B, and 6 have been revised. This revision does not change any conclusions made in this work.
3. In Figures 7, “Brassicaceae” and “Poaceae” have been italicized.
4. In Figure 8, the typo “Oryzae” has been corrected to “Oryza”.

Reviewer 2 Report
I feel the manuscript is overall in good shape. There seems to be a few minor problems in the figures. In Fig.1 the legend should be blue color for Oryzeae. In Fig.8, some genus shades are not labeled. For example, main text mentions each CP subunit has members from both genera.
In the description of supplementary materials, abbreviations are described in Table S2, not S1 as written for Fig. S3 and S4. Title of Table S5 is missing.
The major concern I have is about the method. I know the CTT program is under review separately, but it will still benefit from a brief description of how it works. Based on my understanding, the program starts with Pfam seeds of selected domain families and use blastp to search species protein databases and then use HMMER to validate. Afterwards, the new domain hits are used to search on genome.
I wonder how the RING famlies can be reliably defined by select Pfam families. Additional information like domain organization is used to control false positives for RP and Cullin families. It is necessary because the seed even has AAA families in it. RING Pfam clan also has many RING families in other proteins. Since the threshold for HMMER is pretty loose (E-value 1), I would not be surprised if the search actually finds other RING families as top hit. Did you check to see if irrelevant families are included? Or maybe I missed something for the method.
Author Response
Dear Editor,
In this resubmission, we have summarized below how we have addressed all the comments raised by the reviewer. The reviewer’s comments are copied verbatim in italics. Also, for those paragraphs of the reviewer’s reports that contain comments on unrelated issues, we have broken them down into separate items and addressed each issue separately. In this revision, we also carefully corrected some minor mistakes in Tables S3 and S4 and Figures 2 and 6 in our first submission that have been separately addressed at the end of this letter. These corrections did not change any conclusions made in our work. Please note that unless otherwise stated, all page and figure numbers refer to those of the revised manuscript.
We sincerely thank both referees for their constructive comments, and we believe that having addressed their comments, we have strengthened our manuscript. We hope that you and the referees find our revisions satisfactory.
Response to Reviewer 2
Comment 1:I feel the manuscript is overall in good shape. There seems to be a few minor problems in the figures. In Fig.1 the legend should be blue color for Oryzeae. In Fig.8, some genus shades are not labeled. For example, main text mentions each CP subunit has members from both genera.
Thank you for carefully reading our manuscript. The mistake in Figure 1 has been corrected in the revision.
In Figure 8, only significant genus-specific clades (Statistical significance equal to or greater than 60% of 1,000 bootstrap re-samplings) are shaded. To avoid this confusion, we added one sentence in the figure legend, that reads “Significant clades from Arabidopsisor Oryzagenus only are shaded with light blue or brown color, respectively “, in this revision.
Comment 2:In the description of supplementary materials, abbreviations are described in Table S2, not S1 as written for Fig. S3 and S4. Title of Table S5 is missing.
Thank you for carefully reading our manuscript. This mistake has been corrected in the revision.
Comment 3:The major concern I have is about the method. I know the CTT program is under review separately, but it will still benefit from a brief description of how it works. Based on my understanding, the program starts with Pfam seeds of selected domain families and use blastp to search species protein databases and then use HMMER to validate. Afterwards, the new domain hits are used to search on genome.
We appreciate this comment. This is precisely how the CTT program does. The separate paper about the CTT program has been accepted for publication on PLOS ONE (it has been scheduled for publishing on July 2, 2019). We indeed developed two versions of the program that can be used for a broad range of researchers. Both versions along with detailed installation and running methods are available in two repositories hosted by GitHub at https://github.com/hua-lab/cttand https://github.com/hua-lab/cttdocker. A step-by-step protocol is also available at https://www.protocols.io/view/using-ctt-for-comprehensive-superfamily-gene-annot-zf4f3qw. To make this method clearly, we updated the reference of CTT program and added an additional reference for the CTT step-by-step protocol.
Comment 4:I wonder how the RING famlies can be reliably defined by select Pfam families. Additional information like domain organization is used to control false positives for RP and Cullin families. It is necessary because the seed even has AAA families in it. RING Pfam clan also has many RING families in other proteins. Since the threshold for HMMER is pretty loose (E-value 1), I would not be surprised if the search actually finds other RING families as top hit. Did you check to see if irrelevant families are included? Or maybe I missed something for the method.
We appreciate this comment. The RING E3 family members are the most difficult ones to be annotated primarily due to only few members that have been functionally characterized. To reduce the false discovery as low as possible, we selected the subfamilies within the same SCOOP of RINGv (PF12906, http://pfam.xfam.org/family/RINGv) except for Prok-RING_4, a domain found sporadically in bacteria. The criteria of this selection are that 1) the domain sequences of these subfamilies have been identified in known ubiquitin E3 ligases; 2) they are in the same SCOOP predicted by Pfam; and 3) the number of RING E3 ligases predicted in Arabidopsis thalianain this work is close to the number reported in a previous publication (524 vs. 469; Stone et al., Plant Physiology 2005 (137): 13-30). To make our method clear, we added one sentence that reads “We used the subfamilies in the SCOOP of RINGv predicted to be an E3 ligase on Pfam to annotate RING E3 members (https://pfam.xfam.org; {Bateman, 2007 #60}; Table S1). Prok-RING_4 is excluded because it is found sporadically in bacteria {Burroughs, 2011 #61}” in “Materials and Methods” (Lines 8-11, Page 21).
In order to perform cross-family comparison, we applied the same E-value cutoff 1 for all the 11 families compared in this work. Based on our previous studies on the F-boxand the ubiquitin and ubiquitin-likefamilies (Hua et al., PLOS ONE 2011, Hua et al., Plant J 2018), this cutoff can help identify most, if not all, supferamily members, particularly when the seed sequences are not long, such as RING and F-box families. The effectiveness of CTT annotation and the E-value cutoff 1 can also be exemplified for our annotation of the Skp1 family and the 26S proteasome CP members in A. thalianain this work. Although both families have longer seed sequences than the RING family, we identified the exactly same members of Arabidopsis Skp1 Likegenes and the 26S CP members that have been reported in previous publications (Kong et al., Plant J 2007(50):873-885 and Book et al., JBC 2010(285): 25554-69).
We further checked the E-values of the best RING domains predicted in all RING family proteins in this work. We found 6,570 out of 7,338 (89.5%) predicted members in 14 genomes have an E-value less than 0.001. Therefore, we believe that the E-value cutoff 1 would not cause a significant increase of false discoveries if any.
Additional Changes
1. The number of “Prior Annotation” in Table S3 was not correct. It was a sum of “Prior Annotation” and “New Pep” genes. We have corrected this mistake in this revision. Accordingly, we revised Figure 1B. This revision does not change any conclusions made in this work.
2. The number of loci in Table S4 was not correct. In our first submission, we calculated these numbers without removing differential splicing variants, which resulted in inconsistency between the number of loci listed in this table and the number of sequences collected in File S1 and S2. In this revision, we have carefully removed the splicing variants with short lengths from the same locus. The number of loci in the new Table S4 matches the number of sequences in Files S1 and S2 now. Accordingly, Figures 2A, 2B, and 6 have been revised. This revision does not change any conclusions made in this work.
3. In Figures 7, “Brassicaceae” and “Poaceae” have been italicized.
4. In Figure 8, the typo “Oryzae” has been corrected to “Oryza”.
